# Subcellular Localization of Homomeric TASK3 Channels and Its Presumed Functional Significances in Trigeminal Motoneurons

**DOI:** 10.3390/ijms24010344

**Published:** 2022-12-25

**Authors:** Mitsuru Saito, Chie Tanaka, Hiroki Toyoda, Youngnam Kang

**Affiliations:** 1Department of Oral Physiology, Graduate School of Medical and Dental Sciences, Kagoshima University, Kagoshima 890-8544, Japan; 2Department of Neuroscience and Oral Physiology, Osaka University Graduate School of Dentistry, Suita 565-0871, Japan; 3Department of Behavioral Sciences, Osaka University Graduate School of Human Sciences, Suita 565-0871, Japan

**Keywords:** TASK1, TASK3, homomeric TASK channels, pH sensitivity, cell volume decrease, cGMP

## Abstract

Somatic expressions of either heteromeric TASK1/3 or homomeric TASK1/1 channels have been reported in various neurons, while expression of homomeric TASK3/3 channels has been re-ported only in dendrites. However, it is not known why homomeric TASK3/3 channels are hardly seen in somata of CNS neurons. Given the absence of somatic TASK3/3 channels, it should be clarified why dendritic expression of TASK3/3 channels is inevitable and necessary and how differentially distributed TASK1/1 and TASK3/3 channels play roles in soma-to-dendritic integration. Here, we addressed these questions. We found that TASK3-transfected HEK293 cells showed decreases in cell volume after being transferred from the cultured medium to HEPES Ringer, suggesting that expressions of TASK3 channels in cell bodies cause an osmolarity problem. Using TASK1- and TASK3-transfected oocytes, we also found that cGMP application slightly suppressed TASK3 currents while it largely enhanced TASK1 currents, alleviating the difference between TASK1 and TASK3 currents at physiological pH. As larger motoneurons have extensive dendritic trees while smaller motoneurons have poor ones, cGMP could integrate Ia-EPSPs to recruit small and large motoneurons synchronously by differentially modulating TASKI and TASK3 channels which were complementary distributed in soma and dendrites of motoneurons in the dorsolateral part of the trigeminal motor nucleus.

## 1. Introduction

The resting membrane potential and input resistance (IR) are largely determined by two-pore domain weak inwardly rectifying K^+^ channel (TWIK)-related acid-sensitive K^+^ channel (TASK) 1 and TASK3 [1,2,3]. TASK1 and TASK3 channel subunits can assemble into dimers; homomeric TASK1, homomeric TASK3, and heteromeric TASK1/3 channels [1,4]. The pH sensitivity is different among the three types of dimers [3]. The pH sensitivity of heteromeric TASK1/3 channels is rather similar to that of homomeric TASK1 channels, and their activity at the physiological pH (7.4) is much lower than that at pH 8.4 in contrast to homomeric TASK3 channels [4,5]. The single-channel conductance of homomeric TASK3 or heteromeric TASK1/3 channels is two times larger compared with homomeric TASK1 channels [5].

Heteromeric TASK1/3 channels are expressed in various neurons, such as cerebellar granule cells [5] and hypoglossal motoneurons (MNs) [4]. However, we have recently reported that homomeric TASK1 and TASK3 channels are complementarily distributed in somata and dendrites of MNs in the dorsolateral trigeminal motor nucleus (dl-TMN), respectively [6]. It was also previously reported that homomeric TASK1 channels are expressed in the soma/dendrite of cholinergic neurons in the basal forebrain [7]. Thus, somatic expression of either heteromeric TASK1/3 channels or homomeric TASK1 channels has been reported in various neurons, whereas the somatic expression of homomeric TASK3 channels has not been reported yet until now. Indeed, most CNS neurons usually express both TASK1 and TASK3, except those in striatum/accumbens in which only TASK3 was expressed without TASK1 [8]. However, TWIK1 or other two-pore-domain K^+^ channels are co-expressed with TASK3 to form heterodimeric channels [8,9]. In the present study, we investigated the biological significance of the presence of the homomeric TASK3 channels in the cell body using heterologous expression systems to find the reason why homomeric TASK3 channels are hardly seen in CNS neurons. Given the absence of somatic TASK3 channels, it should also be clarified why dendritic expression of TASK3 channels is inevitable or necessary.

TASK1 and TASK3 subunits have two and one potential cGMP-dependent protein kinase (PKG) phosphorylation sites, respectively [10]. Modulatory action of nitric oxide or cGMP on TASK channels in cholinergic neurons has been reported in many brain regions, such as those in the basal forebrain [11], hypoglossal nucleus [12] and dl-TMN [6,13]. In hypoglossal MNs which express heteromeric TASK1/3 channels, cGMP exerted excitatory effects presumably by inhibiting TASK1/3 channels [12]. In contrast, it is well established that in basal forebrain cholinergic neurons, homomeric TASK1 channels are activated by cGMP [7,11], and it is also reported that homomeric TASK3 channels are inhibited by cGMP [6]. However, it is not known whether and how differentially distributed TASK1 and TASK3 channels in MNs in the dl-TMN are integrated by cGMP to modulate Ia-EPSPs. We have addressed these questions in the present study partly by re-evaluating our previous results.

## 2. Results

### 2.1. Transduction of Homomeric TASK3 Channels

In TASK3-transfected CHO cells (Figure 1A) or HEK293 cells (not shown), we were not able to obtain a stable whole-cell patch-clamp recording with K-gluconate-based internal solution, presumably due to an osmolarity problem caused by large gluconate anion because it was reported to be possible with KCl-based internal solution [5]. Given a large standing outward K^+^ currents flowing through TASK3 channels at a resting condition, a cell volume decrease would occur due to an osmolarity decrease followed by H_2_O outflow [14]. Therefore, we have first examined whether TASK3-transfected HEK293 cells show any cell volume changes following being transferred from the cultured medium to HEPES Ringer solution. In comparison with the control mock-transfected HEK293 cells (Figure 1(C1)) examined immediately after being transferred from the cultured medium to HEPES Ringer solution, TASK3-transfected HEK293 cells showed decreases in the cell volume to be a round/shrunken shape when examined 10 min after being transferred from the cultured medium to HEPES Ringer solution (Figure 1(C2)). In contrast, TASK1-transfected HEK293 cells (Figure 1B) showed no apparent changes in the cell volume and morphology when examined 10 min after being transferred from the cultured medium to HEPES Ringer solution (Figure 1(C3)). Thus, cell volume decreases occurred in TASK3-transfected HEK293 or CHO cells (not shown). Therefore, we have tried to obtain whole-cell recordings before the cell volume largely decreases. Nevertheless, it was still not possible to obtain a stable whole-cell recording from TASK3-expressed cells. The osmolarity problem caused by TASK3 currents (I-TASK3) may have created a secondary osmolarity problem when recorded with K-gluconate-based internal solution (see Discussion). Therefore, using two-electrode voltage clamp method, we next examined I-TASK3 in TASK3-transfected *Xenopus* oocytes, in which it was possible to stably record I-TASK3.

### 2.2. I-TASK3 Recorded from Xenopus Oocyte

I-TASK3 was examined by applying voltage pulses at a holding potential of −90 mV from −150 to +60 mV by 30 mV steps. Following application of 100 μM 8-Br-cGMP, I-TASK3 evoked at pH 7.4 and pH 8.4 was slightly (~7%) decreased (Figure 2(A1,B1)). The pooled data analysis revealed that at pH 7.4, 8-Br-cGMP significantly (*p* = 0.018, two-way ANOVA: drug effect, F(1, 8) = 8.89) decreased I-TASK3 evoked at 0, +30, and +60 mV by 6.4%, 7.0%, and 7.0%, respectively (0 mV, *p* = 0.005; +30 mV, *p* < 0.001; +60 mV, *p* < 0.001, two-way ANOVA followed by Fisher’s PLSD; n = 9) (Figure 2(A2)). Similarly at pH 8.4, 8-Br-cGMP significantly (*p* = 0.001, two-way ANOVA: drug effect, F(1, 8) = 25.36) decreased I-TASK3 evoked at 0, +30, and +60 mV by 7.9%, 7.5%, and 5.9% (0 mV, *p* < 0.001; +30 mV, *p* < 0.001; +60 mV, *p* < 0.001, two-way ANOVA followed by Fisher’s PLSD; n = 9) (Figure 2(B2)). In consistent with the pH sensitivity reported previously [4,5], I-TASK3 was slightly but significantly larger at pH 8.4 than at pH 7.4 (Figure 2(C1); *p* < 0.001, two-way ANOVA: pH effect, F(1, 8) = 95.47). This pH sensitivity was not changed even after I-TASK3 was decreased by 8-Br-cGMP (Figure 2(C2); *p* < 0.001, two-way ANOVA: pH effect, F(1, 8) = 33.98).

### 2.3. I-TASK1 Recorded from Xenopus Oocyte

As it has previously been reported that 8-Br-cGMP enhances I-TASK1 heterologously expressed in HEK293 cells at pH 7.3 [7], we confirmed that 8-Br-cGMP enhances I-TASK1 in *Xenopus* oocytes in which cloned TASK1 channels were heterologously expressed. At pH 7.4, 8-Br-cGMP increased I-TASK1 evoked at 0, +30, and +60 mV significantly (*p* = 0.026, two-way ANOVA: pH effect, F(1, 4) = 12.02) by 75%, 68%, and 60%, respectively (Figure 3(A1,A2); 0 mV, *p* < 0.001; +30 mV, *p* < 0.001; +60 mV, *p* < 0.001, two-way ANOVA followed by Fisher’s PLSD). In contrast, at pH 8.4, 8-Br-cGMP increased I-TASK1 evoked at 0, +30, and +60 mV barely significantly (*p* = 0.049, two-way ANOVA: pH effect, F(1, 4) = 7.84) and only by 16%, 14%, and 11%, respectively (Figure 3(B1,B2); 0 mV, *p* = 0.002; +30 mV, *p* < 0.001; +60 mV, *p* < 0.001, two-way ANOVA followed by Fisher’s PLSD; n = 5). Following increases in pH in the extracellular solution from 7.4 to 8.4, I-TASK1 examined at 0 to +60 was significantly and similarly increased both in the absence and presence of 8-Br-cGMP (Figure 3(C1,C2)). Based on these results, it can be concluded that 8-Br-cGMP slightly or moderately suppresses I-TASK3, whereas it largely enhances I-TASK1.

### 2.4. Integration of Differential Effects of 8-Br-cGMP on TASK1 and TASK3 Channels

Effects of 8-Br-cGMP on the pH sensitivities of I-TASK3 and I-TASK1 evoked at +60 mV in oocytes were examined by measuring the ratios of the amplitudes of I-TASK3 and I-TASK1 recorded at pH 7.4 to those at pH 8.4. The ratio (I_pH7.4_/I_pH8.4_) of I-TASK3 (Figure 4A, Control; 0.88 ± 0.04, n = 9) was significantly larger (*p* < 0.001, unpaired *t*-test: t(12) = −20.65) than that of I-TASK1 (Figure 4B, Control; 0.51 ± 0.02, n = 5), which is consistent with the values reported previously [5]. This difference reflects the difference in the pH-sensitivity between TASK1 and TASK3 as TASK1 is much more sensitive than TASK3 to pH decrease from 8.4 to 7.4 [4]. 8-Br-cGMP application did not significantly (*p* = 0.452, paired *t*-test: t(8) = 0.79) affect the ratio of I-TASK3 (Figure 4A, 8-Br-cGMP; 0.86 ± 0.07, n = 9), whereas it significantly (*p* = 0.031, paired *t*-test: t(4) = −3.27) increased the ratio of I-TASK1 (Figure 4B, 8-Br-cGMP; 0.75 ± 0.17, n = 5). Subsequently, there was no significant difference in the ratio between I-TASK3 and I-TASK1 after 8-Br-cGMP application (Figure 4A,B, 8-Br-cGMP) (*p* = 0.082, unpaired *t*-test: t(12) = −1.90). Additionally, in HEK293 cells in which TASK1 channels were heterologously expressed, it was confirmed that 8-Br-cGMP significantly (*p* = 0.011, paired *t*-test: t(4) = −4.44) increased the ratio of I-TASK1 from 0.57 ± 0.06 to 0.75 ± 0.09 (Figure 4C). These results indicate that 8-Br-cGMP modulated pH sensitivity of TASK1 to increase the amplitude mostly at pH 7.4 while it did not modulate that of TASK3 but significantly decreased the amplitude equally at pH 7.4 and pH 8.4. Subsequently, the difference in the amplitude between I-TASK1 and I-TASK3 at the physiological condition was minimized following an application of 8-Br-cGMP. Consequently, possible differential subcellular distribution of TASK1 and TASK3 channels such as those in trigeminal jaw-closing MNs may be functionally homogenized, and dendritic integration of synaptic inputs may be facilitated.

### 2.5. Subcellular Distributions of TASK1 and TASK3 Channels and Effects of 8-Br-cGMP on Excitatory Postsynaptic Potentials (EPSPs) in MNs in the Dorsolateral Trigeminal Motor Nucleus (dl-TMN)

To confirm the differential subcellular distribution of TASK1 and TASK3 channels in trigeminal jaw-closing MNs, we next re-examined immunoreactivities to anti-TASK1 and anti-TASK3 antibodies in MNs in the dl-TMN. Immunohistochemical stainings revealed that the somata of the ChAT-immunopositive neurons showed strong immunoreactivity to anti-TASK1 antibody (Figure 5A, asterisks), while their proximal and remote dendrites showed relatively weak and virtually no immunoreactivity, respectively (Figure 5A, open and filled arrowheads, respectively). By contrast, the somata of the ChAT-immunopositive neurons showed very faint immunoreactivity to anti-TASK3 antibody (Figure 5B, asterisks), while both the proximal and remote dendrites of these neurons showed a relatively strong immunoreactivity (Figure 5B, open and filled arrowheads, respectively). Thus, these data clearly indicate that TASK1 and TASK3 are almost complementarily expressed in somata and remote dendrites, respectively, although TASK1 and TASK3 were weakly and strongly expressed in the proximal dendrites of MNs, respectively.

It has previously been shown that 8-Br-cGMP largely decreased the IR in smaller MNs with a smaller dendritic domain while it caused either almost no changes or slight increases in the IR in larger MNs with a large dendritic domain due to the differential subcellular distribution of TASK1 and TASK3 channels [6]. Based on the observations of such a differential distribution of TASK1 and TASK3 channels and the differential effects of 8-Br-cGMP on TASK1 and TASK3 channels, it can be assumed that the presumed Ia-EPSPs would be modulated by 8-Br-cGMP depending on whether the relevant synapse is located on the soma, proximal or remote dendrites and whether in smaller or larger MNs. To address this possibility, we next re-examined and re-analyzed the effects of 8-Br-cGMP on the presumed Ia-EPSPs evoked in MNs in the dl-TMN by microstimulation applied just dorsal to the TMN. The pooled data analysis of 16 MNs revealed that application of 100 μM 8-Br-cGMP significantly increased the amplitude of EPSPs evoked in MNs in spite of the large increase in I-TASK1 (21.9 ± 28.8%, *p* = 0.008, paired *t*-test: t(15) = −3.05) while it either increased or decreased IR in MNs with no significant change in their mean value (−3.0 ± 8.0%, *p* = 0.150, paired *t*-test: t(15) = 1.52), reflecting the dispersion in the soma and dendritic domain sizes (Figure 5C,D). Furthermore, there was a significant but moderate positive correlation between the changes in EPSP amplitude and those in IR caused by application of 8-Br-cGMP (Figure 5D; *p* = 0.025, r = 0.56, Pearson’s correlation). These results strongly suggest that both the electrotonic distance from the soma to the synaptic site and the cell size affected the changes in EPSP amplitude caused by 8-Br-cGMP application, because the location of the synapses responsible for evoking these EPSPs largely varied mostly along proximal to remote dendrites of MNs in the dl-TMN [15,16] (see Discussion).

## 3. Discussion

In the present study, a cell volume decrease was caused only in TASK3-expressed HEK293 cells (Figure 1C,D). This volume decrease may have been caused by a large standing outward K^+^ currents flowing through TASK3 channels at a resting condition [17]. A similar contribution of outward K^+^ currents to cell volume decrease has been reported in glioma cells, in which the regulatory volume decrease by outward K^+^ currents occurred against the cell swelling induced by an osmolarity reduction in the extracellular solution [14]. Such a cell volume decrease may further lead to cell death unless compensated with the activities of Na^+^, K^+^-2Cl^−^ cotransporter (NKCC1) [18] and Na^+^-K^+^-pump [19] while subsequently increased Cl^-^ anion must be ejected by K^+^-Cl^−^ cotransporter 2 (KCC2) or volume regulated anion channel (VRAC) [20,21,22]. This compensation mechanism appeared to be hampered when whole-cell recording was performed with K-gluconate based internal solution, as encountered in the present study but not with KCl-based [5] or KCH_3_SO_3_-based internal solution [4]. This may be because relatively larger anion such as gluconate^-^ may have blocked VRAC as reported previously [23] while smaller anion such as Cl^−^ or CH_3_SO_3_^−^ can flow VRAC [24]. Thus, homomeric TASK3 channels may have caused a secondary osmolality problem when recorded with K-gluconate based internal solution. Because mammalian neurons contain large protein anions intracellularly, homomeric TASK3 channels may not be expressed in somata of CNS neurons. Indeed, in a previous study [8], it was demonstrated that expression of TASK3 was almost always co-expressed with TASK1 in most CNS neurons except those in striatum/accumbens where TASK3 was co-expressed with TREK1 or other two-pore-domain K^+^ channels instead of TASK1 to form heterodimeric channels [8,9]. On the other hand, the functional significance of dendritically expressed TASK3 channels in trigeminal MNs is clear (Figure 5). Dendritic expression of homomeric TASK3 channels may limit the amount of K^+^ outward current not to lead to the cell volume decrease because even with a small TASK3 currents the membrane potential in the dendrites is easily hyperpolarized to the level of K^+^ reversal potential due to high input resistance of dendrites [25]. This would be the reason why homomeric TASK3 channels are expressed mostly in dendrites instead of the cell body.

It is very interesting that TASK1 and TASK3 are oppositely modulated by 8-Br-cGMP although their molecular mechanisms are not known yet. Optimum PKG phosphorylation sites of TASK1 and TASK3 channels are S409 and S394 located in C-terminal of TASK channels, respectively [10] while pH sensitive sites are H98 and K210/D211 for TASK1 channels and only H98 for TASK3 channels [7,26,27]. In the present study, PKG increased the amplitude of I-TASK1 at pH 7.4 by changing the pH sensitivity in TASK1 channels (Figure 3 and Figure 4), while PKG decreased the amplitude of I-TASK3 without changing pH sensitivity (Figure 2 and Figure 4). The pH sensitive site of K210/D211 for TASK1 channels is located between the second pore-helix and the inner-helix M4, which can be easily accessed by PKG-phosphorylation site of C-terminal (S409) [27]. In contrast, in TASK3 channels, the sole pH sensitive site of H98 located between the first pore-helix and the inner-helix M2 is remote from the PKG-phosphorylation site of C-terminal (S394) [28]. Thus, pH sensitivity of TASK1 channels may be modulated by PKG while gating of TASK3 channels may be modulated by PKG.

In our previous study, there were significant and strong negative correlations between IRs and changes in IR caused by 8-Br-cGMP application and also between IRs and changes in EPSP amplitudes caused by 8-Br-cGMP application [6]. In the present study, however, there was a significant but only moderate positive correlation between the changes in IR and those in EPSP amplitude caused by application of 8-Br-cGMP (Figure 5D). Because synaptic locations of Ia afferents are distributed along the proximal to remote dendrites of MNs in dl-TMN [15,16], the electrotonic distance from the soma to the respective synaptic sites would be dispersed as were the case with the soma and dendritic domain sizes. It should also be noted that this moderate but not strong positive correlation may also be due to the possible contamination of Ia inputs with others activated in response to stimulation applied just dorsal to the TMN [6,29].

The present study also revealed that proximal dendrites were weakly and moderately to strongly immunopositive for TASK1 and TASK3, respectively, while the remote dendrites were immunonegative and strongly immunopositive for TASK1 and TASK3, respectively (Figure 5A,B). Therefore, the difference in the electrotonic distance between the synaptic inputs onto the proximal dendrites and those onto the remote dendrites would be reduced by 8-Br-cGMP mainly through the inhibition of I-TASK3, and subsequently the integrated EPSPs amplitude would increase (Figure 5C). However, IRs were either decreased or increased by 8-Br-cGMP application. As IRs were increased more due to the inhibition of TASK3 as expressed in many proximal dendrites of large MNs, the integrated EPSPs tended to increase more in amplitude. In contrast, the increase in the integrated EPSPs amplitude tended to become smaller or even turned into a decrease as IRs were decreased more due to the poor proximal dendrites as in smaller MNs. Thus, activation of cGMP pathway integrated Ia-synaptic inputs on the proximal/remote dendrites through the inhibition of TASK3 and also modulated soma-dendritic integration through opposing modulation of TASK1 and TASK3 channels in trigeminal jaw-closing MN, subsequently inducing a synchronous activation of small and large MNs.

In conclusion, homomeric TASK3 channels are not likely expressed in somata of mammalian neurons due to the osmolarity problem. The cGMP facilitates TASK1 but inhibits TASK3 currents, thereby decreasing the difference between TASK1 and TASK3 currents at the physiological pH condition. Subsequently, cGMP can integrate Ia-EPSPs to recruit small and large MNs synchronously by differentially modulating TASKI and TASK3 channels which were complementary distributed in soma and dendrites of MNs in the dl-TMN. Elucidation of these modulations may lead to a further elucidation of the mechanism for the modulation of the rank order recruitment of motor units [30] as larger MNs have extensive dendritic trees while smaller MNs have poor dendritic trees.

## 4. Materials and Methods

Animal experiments were performed in accordance with the guidelines for the use of laboratory animals of Osaka University Graduate School of Dentistry. The experimental protocol was approved by the Institutional Animal Care and Use Committee at Osaka University Graduate School of Dentistry (Approval No. 2873).

### 4.1. Whole-Cell Patch-Clamp Recordings from Cultured Mammalian Cells

We examined TASK3 currents (I-TASK3) heterologously expressed in HEK293 cells (human embryonic kidney cells 293; EC85120602-F0, KAC Co., Ltd., Osaka, Japan) and CHO cells (Chinese hamster ovary cells; EC85050302-F0, KAC). Mouse TASK3 cDNAs were used [6]. Lipofectamine LTX (ThermoFisher, Rockford, IL, USA) was used for plasmid transfection experiments according to the manufacturer’s protocol. Briefly, Opti-MEM (Thermo Fisher Scientific, Waltham, MA, USA) that contained pIRES2-DsRed2-TASK3 cDNA and Lipofectamine Plus reagent (ThermoFisher) were mixed and incubated for 5 min, added to LTX reagent (ThermoFisher), and kept for 30 min at room temperature. This solution was added stepwise to cells and gently mixed. Cells were incubated at 37 °C in 5% CO_2_ for 3 to 6 h and plated onto glass coverslips after the medium was replaced with fresh complete medium. The fresh complete medium for HEK293 contained DMEM (Dulbecco’s modified Eagle’s medium, ThermoFisher) supplemented with 10% FBS (fetal bovine serum, ThermoFisher), 100 U/mL penicillin (ThermoFisher) and 100 μg/mL streptomycin (ThermoFisher), and that for CHO contained Ham’s F-12 Nutrient Mixture (ThermoFisher) supplemented with 10% FBS, 100 U/mL penicillin and 100 μg/mL streptomycin.

One day after transfection, cells were plated onto glass coverslips. Electrophysiological recordings were performed 36–60 h after transfection. Red fluorescence from DsRed2-expressing cells was identified using a microscope (BX-51WI IR-DIC/FL; Olympus, Tokyo, Japan) equipped with excitation (554 nm) and barrier filters (591 nm) and a mercury lamp light source. Axopatch 200B (MDS Analytical Technologies, Sunnyvale, CA, USA) was used for the experiments. The extracellular solution had the following composition: 136 mM NaCl, 1.8 mM KCl, 2.5 mM CaCl_2_, 1.3 mM MgCl_2_, 10 mM HEPES, 1.2 mM KH_2_PO_4_, 10 mM glucose, and the pH 7.4/8.4 was adjusted using NaOH. The internal solution had the following composition (in mM): 123 mM K-gluconate, 8 mM KCl, 20 mM NaCl, 0.5 mM MgCl_2_, 0.5 mM ATPNa_2_, 0.3 mM GTP-Na_3_, 10 mM HEPES, 0.1 mM EGTA; the pH was adjusted to 7.3 with KOH. The patch pipettes had a DC resistance of 4–5 MΩ when filled with the internal solution. Recordings were made at room temperature. The sealing resistance was usually >10 GΩ. Whole-cell currents were low pass filtered at 2 kHz (four-pole Bessel filter), digitized at a sampling rate of 2−10 kHz (Digidata 1322A, MDS Analytical Technologies). Under the voltage-clamp condition at the holding potential of −70 mV, depolarizing ramp (−130 to −50 mV, 0.9 s duration) pulses were applied alternately every 10 s. The current amplitude between −95 and −70 mV was measured. The digital images were captured using a CCD camera (C5999; Hamamatsu Photonics, Hamamatsu, Japan).

### 4.2. Two-Electrode Voltage-Clamp Recordings from Xenopus Oocytes

The cloned TASK1 and TASK3 channels were heterologously expressed in *Xenopus* oocytes. The isolation and maintenance of the oocytes of frogs (*Xenopus* laevis) and injection with cRNA were performed as described previously [31]. Mouse TASK1 and TASK3 cDNAs were used [6,7]. The cRNAs for injection into oocytes were prepared with T7 RNA polymerase (ThermoFisher). After oocytes were injected with cRNAs for TASK1 (50 ng/oocyte) and TASK3 (50 ng/oocyte), these oocytes were incubated at 18 °C in ND96 solution that contained the following: 96 mM NaCl, 2 mM KCl, 1.8 mM CaCl_2_, 1 mM MgCl_2_, and 5 mM HEPES, pH 7.4 with NaOH, and supplemented with gentamicin (50 μg/mL). Currents were recorded 1–3 days after the cRNA injection.

Two electrode voltage clamp was performed to record membrane currents of oocytes using the GeneClamp 500 amplifier (MDS) at room temperature. Data were analyzed with pCLAMP version 10 and Clampfit version 10.2 (MDS). The bath solution contained the following: 96 mM NaCl, 2 mM KCl, 1.8 mM CaCl_2_, 1 mM MgCl_2_, and 5 mM HEPES, pH 7.4 or pH 8.4 was adjusted with NaOH. The tip resistance of the glass electrodes was 0.4–1.5 MΩ when filled with the 3 M KCl pipette solution. Niflumic acid (0.1 mM; Sigma-Aldrich) was added in the bath solution to block Ca^2+^-dependent Cl^−^ channels that are richly expressed in the plasma membrane of oocytes [32]. 8-Br-cGMP was bath applied at concentration of 100 μM. TASK currents (I-TASK) were evoked by voltage steps ranging between –150 and +60 mV in 30 mV steps applied at a holding potential of –90 mV. The currents were measured 5–10 ms before the offset of the voltage step.

### 4.3. Whole-Cell Recording from Motoneurons (MNs) in the Dorsolateral Trigeminal Nucleus (dl-TMN)

The procedure for slice preparation was the same as that in the previous study [6]. Using Wistar rats of both sexes at postnatal days 7–14 (Nihon Dobutsu, Osaka, Japan), coronal sections of 250 μm thickness including the dl-TMN were cut. The electrophysiological studies were performed on the MNs in the dl-TMN. Axopatch 200B (MDS Analytical Technologies) was used for whole-cell patch-clamp recordings. The extracellular solution had the following composition: 124 mM NaCl, 1.8 mM KCl, 2.5 mM CaCl_2_, 1.3 mM MgCl_2_, 26 mM NaHCO_3_, 1.2 mM KH_2_PO_4_, and 10 mM glucose, bubbled with mixture of 95% O_2–_5% CO_2_. The internal solution had the following composition: 123 mM K-gluconate, 8 mM KCl, 20 mM NaCl, 0.5 mM MgCl_2_, 2 mM ATP-Na_2_, 0.3 mM GTP-Na_3_, 10 mM HEPES, and 0.1 mM EGTA; the pH was adjusted to 7.3 with KOH. The patch pipettes had a DC resistance of 4–5 MΩ when filled with the internal solution. The membrane potential values were corrected for the junction potential between the internal solution and the standard extracellular solution (–10 mV). All recordings were made at room temperature. The sealing resistance was more than 10 GΩ. Whole-cell currents or voltages were low-pass filtered at 2 kHz (4-pole Bessel filter), digitized at a sampling rate of 10 kHz (Digidata 1322A, MDS Analytical Technologies). Microstimulation of 100 μs duration was delivered via a sharp monopolar tungsten electrode (DC resistance, 1 MΩ), which was placed just dorsal to the TMN to simulate the presumed Ia input arising from the mesencephalic trigeminal sensory neurons [29]. The intensity of stimulus current was lower than 5 μA. 8-Br-cGMP was bath-applied at concentration of 100 μM.

### 4.4. Fluorescence Immunohistochemistry

For immunostaining, three Wistar rats were used for the preparation of fixed brains as described previously [6]. Coronal sections at 40 μm thickness of the fixed brains were incubated overnight in phosphate-buffered saline containing Triton X-100, λ-carrageenan and donkey serum with anti-[rat TASK1] guinea pig antibody (2 μg/mL) or anti-[rat TASK3] rabbit antibody (1 μg/mL) (Alomone Labs, Jerusalem, Israel) with or without anti-ChAT goat antiserum (1:500; AB144P, Chemicon, Temecula, CA, USA). After a wash with PBS, the sections were incubated for 2 h with biotinylated anti-[guinea pig IgG] donkey antibody (10 μg/mL) (Jackson ImmunoResearch, West Grove, PA, USA) or biotinylated anti-[rabbit IgG] donkey antibody (10 μg/mL) (Jackson ImmunoResearch) and then incubated for 1 h with ABC in phosphate-buffered saline containing Triton X-100. For the immunostaining of TASK1, the sections were then incubated for 30 min with BT-GO reaction mixture and was followed by incubation for 1 h with Alexa Fluor 594-conjugated streptavidin (5 μg/mL). For the immunostainings of TASK3, the sections were then incubated for 30 min with TSA Cyanine 3 (Cy3) System (PerkinElmer, Waltham, MA, USA). For the immunostainings of ChAT, the sections were then incubated for 2 h with Alexa Fluor 488-conjugated anti-[goat IgG] donkey antibody (10 μg/mL) (Invitrogen) and Cy3-conjugated streptavidin (Invitrogen) in the presence of 10% normal rabbit serum. The sections were observed with a confocal laser-scanning microscope (LSM510, Zeiss, Oberkochen, Germany). Alexa Fluor 488 and 594 were excited with 488 nm and 543 nm laser beams, and observed through 505–530 nm and >560 nm emission filters, respectively. The digital images were captured by using a software (LSM510).

### 4.5. Data Analysis

Statistical analysis was performed using STATISTICA10J (StatSoft). Numerical data were expressed as the mean ± SD. The statistical significance was assessed using paired or unpaired Student’s *t*-test and one way or two-way ANOVA followed by post hoc Fisher’s protected least significant difference (PLSD) test. Statistical results are given as a *p* value, except when it is very small (*p* < 0.001). *p* < 0.05 was considered statistically significant.

## Figures and Tables

**Figure 1 ijms-24-00344-f001:**
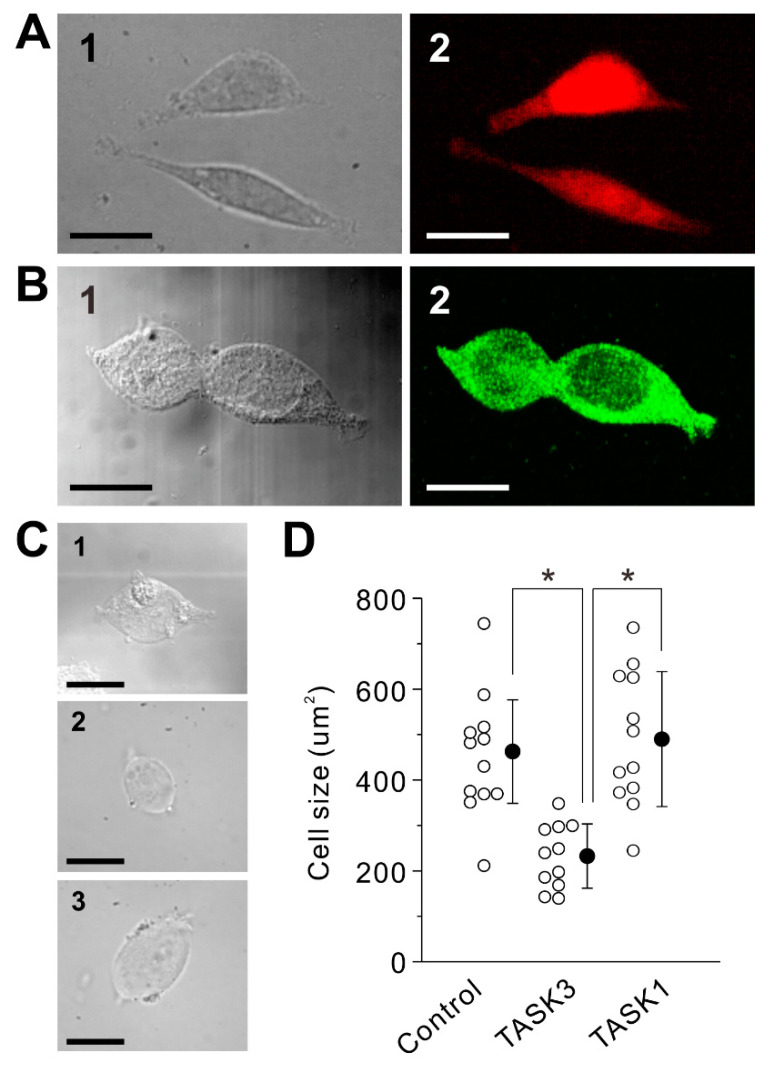
Morphological changes of TASK1- and TASK3- transfected mammalian cells in HEPES Ringer solution. (**A**) An infrared differential interference contrast (IR-DIC) image (**1**) and a DsRed2 fluorescent image (**2**) of TASK3-transfected CHO cells, captured immediately after being transferred from the cultured medium to HEPES Ringer solution. Scale bar, 20 μm. (**B**) An IR-DIC image (**1**) and a ZsGreen1 fluorescent image (**2**) of TASK1-transfected HEK293 cells, captured immediately after being transferred from the cultured medium to HEPES Ringer solution. Scale bar, 20 μm. (**C**) An IR-DIC image of a control mock-transfected HEK293 cell (**1**), captured immediately after being transferred from the cultured medium to HEPES Ringer solution and those of IR-DIC images of TASK3- (**2**) and TASK1-transfected HEK293 cells (**3**), captured 10 min after being transferred from the cultured medium to HEPES Ringer solution. Scale bar, 20 μm. (**D**) Cell sizes of nontransfected HEK293 cells which were taken just after being transferred from the cultured medium to HEPES Ringer solution and those of TASK3- and TASK1-transfected HEK293 cells which were taken 10 min after being transferred from the cultured medium to HEPES Ringer solution. Hollow and filled circles represent the individual and mean values of cell size, respectively. * *p* < 0.001, one-way ANOVA followed by Fisher’s PLSD.

**Figure 2 ijms-24-00344-f002:**
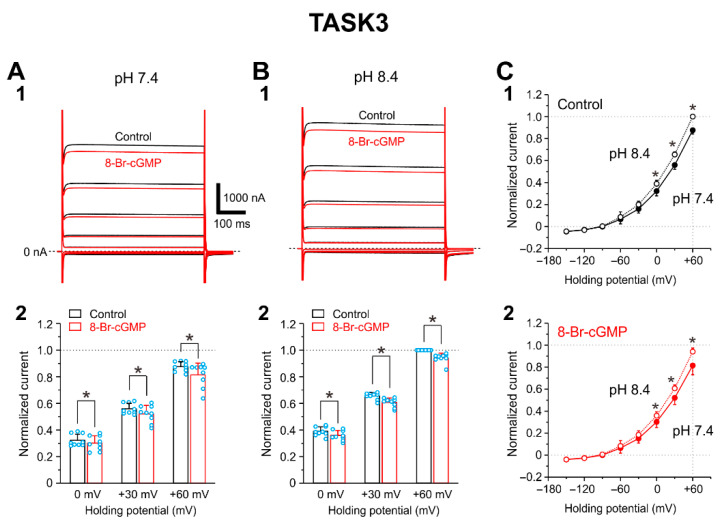
8-Br-cGMP inhibits cloned TASK3 channels expressed in oocytes. (**A1**,**B1**) Superimposed traces of I-TASK3 obtained evoked by voltage steps ranging between −150 and +60 mV in 30 mV steps applied at a holding potential of −90 mV at pH 7.4 (**A1**) and 8.4 (**B1**) obtained before (black traces) and after applying 100 μM 8-Br-cGMP (red traces). (**A2**,**B2**) The normalized I-TASK3 recorded before and during the application of 8-Br-cGMP at pH 7.4 (**A2**) and those at pH 8.4 (**B2**). All the current amplitudes were normalized by the amplitude of I-TASK3 evoked at +60 mV in the extracellular solution at pH 8.4. * *p* < 0.05, two-way ANOVA followed by Fisher’s PLSD. (**C1**,**C2**) I-V relationship of I-TASK3 examined at pH 8.4 and pH 7.4 in the absence (**C1**) and presence of 8-Br-cGMP (**C2**). Following increases in pH in the extracellular solution from 7.4 to 8.4, I-TASK3 examined at 0 to +60 was significantly but not prominently increased both in the absence and presence of 8-Br-cGMP. Hollow and filled circles represent the mean amplitudes of normalized currents obtained at pH 8.4 and 7.4, respectively. * *p* < 0.05, two-way ANOVA followed by Fisher’s PLSD.

**Figure 3 ijms-24-00344-f003:**
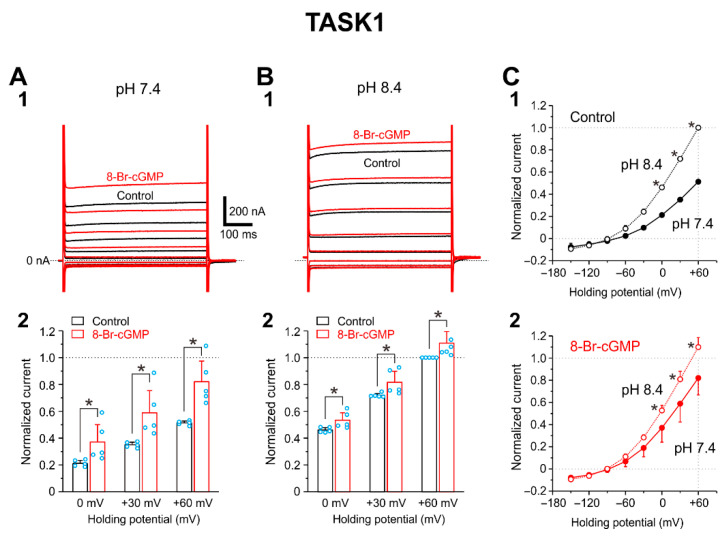
8-Br-cGMP enhances heterologously expressed TASK1 channels in oocytes. (**A1**,**B1**) Superimposed traces of I-TASK1 evoked by voltage steps ranging between −150 and +60 mV in 30 mV steps applied at a holding potential of −90 mV at pH 7.4 (**A1**) and pH 8.4 (**B1**) obtained before (black traces) and after applying 100 μM 8-Br-cGMP (red traces). (**A2**,**B2**) The normalized values of I-TASK1 recorded before and during the application of 8-Br-cGMP at pH 7.4 (**A2**) and those at pH 8.4 (**B2**). All the current amplitudes were normalized by the amplitude of I-TASK1 evoked at +60 mV in the extracellular solution at pH 8.4. * *p* < 0.05, two-way ANOVA followed by Fisher’s PLSD. (**C1**,**C2**) I-V relationship of I-TASK1 examined at pH 8.4 and pH 7.4 in the absence (**C1**) and presence of 8-Br-cGMP (**C2**). Following increases in pH in the extracellular solution from 7.4 to 8.4, I-TASK1 examined at 0 to +60 was significantly and similarly increased both in the absence and presence of 8-Br-cGMP. Hollow and filled circles represent the mean amplitudes of normalized currents obtained at pH 8.4 and 7.4, respectively. * *p* < 0.05, two-way ANOVA followed by Fisher’s PLSD.

**Figure 4 ijms-24-00344-f004:**
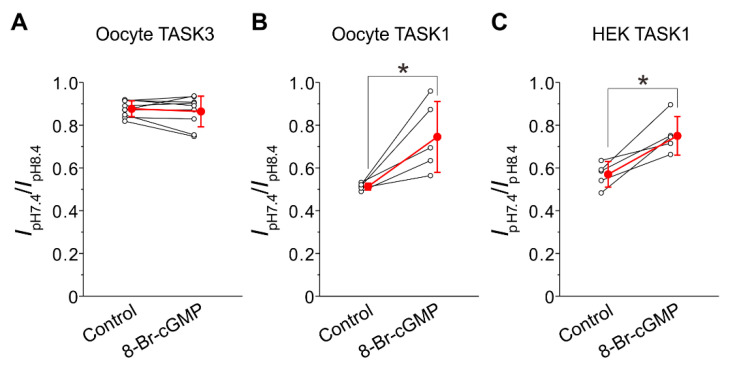
Differential effects of 8-Br-cGMP on TASK3 and TASK1 channels. (**A**,**B**) The ratio of the I-TASK3 amplitude at pH 7.4 to that at pH 8.4 recorded at the membrane potential of +60 mV in oocytes obtained before and following 8-Br-cGMP application in oocytes (**A**) and that of the I-TASK1 amplitude (**B**). Note that 8-Br-cGMP significantly increased the ratio of I-TASK1 (**B**), whereas it did not significantly affect the ratio of I-TASK3 (**A**), consequently largely decreasing the difference in the ratio between TASK1 and TASK3. * *p* < 0.05, paired *t*-test. (**C**) The ratio of the I-TASK amplitude at pH 7.4 to that at pH 8.4 recorded between −95 and −70 mV obtained before and following 8-Br-cGMP application in HEK293 cells. Note that 8-Br-cGMP significantly increased the ratio of I-TASK1 in HEK293 cells as well as in oocytes. Hollow and filled circles represent the individual and mean values of the normalized current amplitudes, respectively. Red line represents the change in the mean current amplitude following application of 8-Br-cGMP. * *p* < 0.05, paired *t*-test.

**Figure 5 ijms-24-00344-f005:**
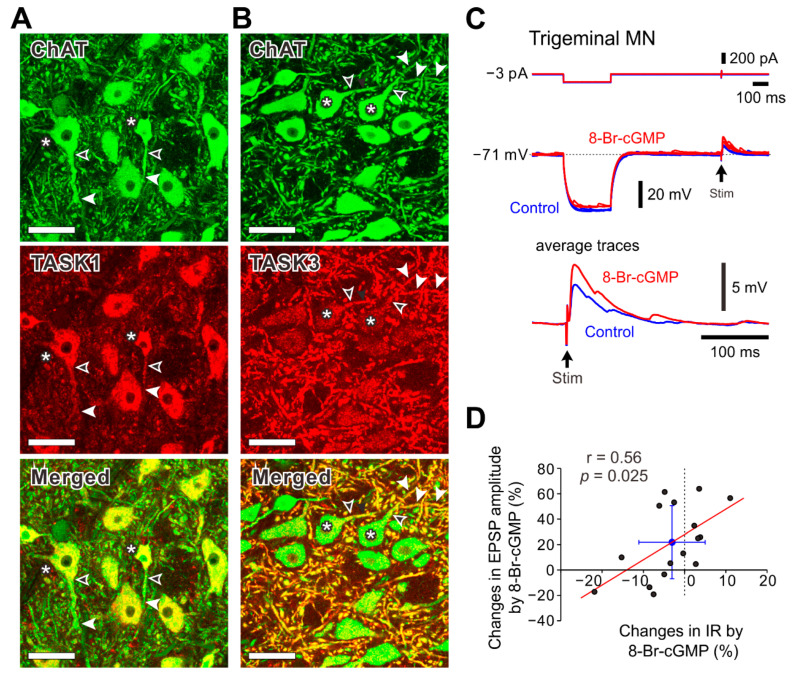
Distribution of TASK1 and TASK3 channels and effects of 8-Br-GMP on EPSPs in MNs in the dl-TMN. (**A**) Confocal micrographs showing immunoreactivity to anti-ChAT antibody (green) and anti-TASK1 antibody (red) in MNs in the dl-TMN. The somata of the ChAT-immunopositive neurons showing strong immunoreactivity to anti-TASK1 antibody (asterisks), while their proximal and remote dendrites showing relatively weak and virtually no immunoreactivity, respectively (open and filled arrowheads). Scale bars: 50 μm. (**B**) Confocal micrographs showing immunoreactivitiy to anti-ChAT antibody (green) and anti-TASK3 antibody (red) in MNs in the dl-TMN. The somata of the ChAT-immunopositive neurons showing very faint immunoreactivity to anti-TASK3 antibody (asterisks), while the proximal and remote dendrites of these neurons showed relatively weak and strong immunoreactivity, respectively (open and filled arrowheads). Scale bars: 50 μm. (**C**) EPSPs evoked by stimulation applied just dorsal to the TMN obtained before and during application of 8-Br-cGMP in a MN in the dl-TMN. Application of 8-Br-cGMP increased the amplitude of EPSPs evoked in MNs while it decreased IR. (**D**) A significant but moderate positive correlation between changes in EPSP amplitude and those in IR following application of 8-Br-cGMP (n = 16). Red line represents a regression line and the blue lines represent the SD values for changes in amplitude and IR.

## Data Availability

The data presented in this study are available in request from the corresponding author.

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
