# Peer review of "Subcellular Localization of Homomeric TASK3 Channels and Its Presumed Functional Significances in Trigeminal Motoneurons"

_ijms, 2022, doi:10.3390/ijms24010344_

Round 1

Reviewer 1 Report

The submitted manuscript tried to reveal the functional significances of TASK3/3 homomeric channels in trigeminal motoneurons through the detection of its subcellular localization and investigating the effects of 8-Br-GMP on EPSPs. Although it provided some interesting founding and was well organized, it needs to be made a major revision before acceptance.

1 The abstract is not well organized. The background was a relatively longer description, while the major results were not clearly presented and the conclusion of the current study was not addresses.  

2 Figure 1. Figure 1A is the images of TASK1 transfection in CHO cells. However, 1D is the summarized data of the untransfected and transfected HEK293 cells. In addition, 1C should provide the infrared differential interference contrast (IR-DIC) image and fluorescent image of control and TASK1- and TASK3- expressing cell at the same time, even in the same filed.

3 The manuscript should present a conclusion section.

Author Response

Authors’ responses to the comments made by Reviewer #1

Thank you very much for your constructive comments on how to improve our manuscript. Our responses to your comments are as follows:

  1. The abstract is not well organized. The background was a relatively longer description, while the major results were not clearly presented and the conclusion of the current study was not addresses.

Response: Thank you for the suggestion. We have extensively revised the abstract. We have shortened the background while we have clearly described the major results and added the conclusion in the revised manuscript.

  1. Figure 1. Figure 1A is the images of TASK1 transfection in CHO cells. However, 1D is the summarized data of the untransfected and transfected HEK293 cells. In addition, 1C should provide the infrared differential interference contrast (IR-DIC) image and fluorescent image of control and TASK1- and TASK3-expressing cell at the same time, even in the same filed.

Response: Thank you for the suggestion. We have added the IR-DIC image of a control mock-transfected HEK cell which was taken immediately after being transferred from the cultured medium to HEPES Ringer solution as shown in the new Fig. 1C1. However, we did not obtain the fluorescent images of these cells because the sizes of TASK1- and TASK3-expressing cells must be compared with that of non-fluorescent IR-DIC image of control mock-transfected cells.

  1. The manuscript should present a conclusion section.

Response: We have added a conclusion section (lines 324-332).

Reviewer 2 Report

Subcellular localization of TASK3/3 homomeric channels and 2 its presumed functional significances in trigeminal 3 motoneurons

Study regarding Subcellular localization of TASK3/3 homomeric channels and 2 its presumed functional significances in trigeminal 3 motoneurons, addressed well by the authors with the help of suitable experiments. It was nice to read this article throughout. My major concern is related to discussion part. Which doesn’t justify the outcome of the results and authors can think about rewriting this section mainly based on what they got from the results. Moreover, results are super descriptive so it’s easy to get information regarding the output of the work. A few of my suggestions are given below:

Introduction

Introduction is written clearly and precisely.

I am not convinced with the aim of the study (line 55), just because it’s hardly seen in the CNS?

If authors can find out what it’s biological/ clinical significance, then it will impact on the study much more than now.

Results: Major part of lines 72-79 and 86-90 should be described in discussion part instead

Is there any reason behind the missing scale bar in the TASK3 transfected HEK293 cells? (FIG1 C2)

Fig 1: line 103-106, description will go into the result section.

Fig 5: For better clarity, it will be nice to move the asterisk aside than just in the center of the neuron.

Discussion: Discussion part needs to be rewrite mainly first two paragraphs where authors go beyond their experimental output. Even most of them are missing references (I mentioned a few of them below).

Line 252-253: Needs to be justified by a suitable reference.

Line 262-265: Needs to be justified by a suitable reference.

Line 266-274: Needs to be justified by a suitable reference.

In result section, authors mention so many justifications for their experiment, in my suggestion it’s better to move them in discussion part and just keep the discussion to the point of outcome of results.

Moreover, in the last paragraph, the question raised in the study was discussed nicely. It will be better to mention future directions and limitations of the study. 

Author Response

Authors’ responses to the comments made by Reviewer #2

Thank you very much for your constructive comments on how to improve our manuscript. Our responses to your comments are as follows:

Introduction

  1. I am not convinced with the aim of the study (line 55), just because it’s hardly seen in the CNS? If authors can find out what it’s biological/clinical significance, then it will impact on the study much more than now.

Response: In response to this criticism, we have revised the manuscript as follows: “we investigated the biological significance of the presence of the homomeric TASK3 channels in the cell body using heterologous expression systems to find the reason why homomeric TASK3 channels are hardly seen in CNS neurons” (lines 54-56).

  1. Results: Major part of lines 72-79 and 86-90 should be described in discussion part instead.

Response: We have revised and shortened the descriptions in lines 72-79, while we left the core description to give a rationale for the osmolarity experiment in the result section (see lines 75-78 in the revised manuscript). We have moved the descriptions in lines 86-90 into the discussion (lines 261-264 in the revised manuscript).

  1. Is there any reason behind the missing scale bar in the TASK3 transfected HEK293 cells? (FIG1 C2).

Response: We have added the scale bars in the new Fig. 1C2 and Fig. 1C3.

  1. Fig 1: line 103-106, description will go into the result section.

Response: We have described the result section more in detail by partly using the description in lines 103-106 as follows: “In comparison with the control mock-transfected HEK293 cells (Fig. 1C1) examined immediately after being transferred from the cultured medium to HEPES Ringer solution, TASK3-transfected HEK293 cells showed decreases in the cell volume to be a round/shrunken shape when examined 10 min after being transferred from the cultured medium to HEPES Ringer solution (Fig. 1C2). In contrast, TASK1-transfected HEK293 cells (Fig. 1B) showed no apparent changes in the cell volume and morphology when examined 10 min after being transferred from the cultured medium to HEPES Ringer solution (Fig. 1C3)" (lines 81-88 in the revised manuscript).

  1. Fig 5: For better clarity, it will be nice to move the asterisk aside than just in the center of the neuron.

Response: Following this suggestion, we have placed the asterisks aside the neuron in Fig. 5A, in which ChAT and TASK1 immunoreactivities were examined. However, we maintained the position of asterisks in Fig. 5B, in which ChAT and TASK3 immunoreactivities were examined. This is because TASK3 immunoreactivity was very weak in the soma. If we placed an asterisk aside the soma, it would be very unclear what is indicated by the asterisk.

  1. Discussion: Discussion part needs to be rewrite mainly first two paragraphs where authors go beyond their experimental output. Even most of them are missing references (I mentioned a few of them below). Line 252-253: Needs to be justified by a suitable reference. Line 262-265: Needs to be justified by a suitable reference. Line 266-274: Needs to be justified by a suitable reference.

Response: We have extensively revised the first two paragraphs by referring to the experimental results (lines 255-282), and also added the missing reference for the description in lines 261-264. Regarding the description in lines 268-271, however, we just discussed the behavior of TASK3 current flowing at dendrites with high input resistance, which can be expected as a theoretical consequence.

  1. In result section, authors mention so many justifications for their experiment, in my suggestion it’s better to move them in discussion part and just keep the discussion to the point of outcome of results.

Response: Following this comment, we have moved some descriptions that justify our experiments or experimental results into the discussion. However, we maintained some of the descriptions to give a rationale for the respective experiments and we kept the discussion to the point of outcome of results in the revised manuscript.

  1. Moreover, in the last paragraph, the question raised in the study was discussed nicely. It will be better to mention future directions and limitations of the study.

Response: We have described the future direction of the study to clarify the mechanism for the rank order recruitment of motor units as larger MNs have extensive dendritic trees while smaller MNs have poor dendritic trees (lines 330-332).

Regarding the limitation of our study, as the electrical stimulation was applied dorsal to the TMN to simulate the presumed Ia input arising from the mesencephalic trigeminal sensory neurons (Shigenaga et al, 1988, Brain Res.), we have described the limitation in the discussion (lines 305-307) as follows: “It should also be noted that this moderate but not strong positive correlation may also be due to the possible contamination of Ia inputs with others activated in response to stimulation applied just dorsal to the TMN [6, 29].”

Round 2

Reviewer 1 Report

 The authors have made major revisions based on the reviewers' comments, and the quality of the revised manuscript has been greatly improved. I am satisfied with the response to the comments, and recommend that it can be accepted after a minor revision of the references format consistent with the style of this journal.

Author Response

We have corrected the references format to be consistent with the style of this journal.